# Histone H4K20 methylation mediated chromatin compaction threshold ensures genome integrity by limiting DNA replication licensing

Muhammad Shoaib [1], David Walter[1], Peter J. Gillespie[2], Fanny Izard[3], Birthe Fahrenkrog [4], David Lleres[5], Mads Lerdrup[1], Jens Vilstrup Johansen[1], Klaus Hansen[1], Eric Julien [3,6], J. Julian Blow [2] & Claus S. Sørensen [1]

The decompaction and re-establishment of chromatin organization immediately after mitosis is essential for genome regulation. Mechanisms underlying chromatin structure control in daughter cells are not fully understood. Here we show that a chromatin compaction threshold in cells exiting mitosis ensures genome integrity by limiting replication licensing in G1 phase. Upon mitotic exit, chromatin relaxation is controlled by SET8-dependent methylation of histone H4 on lysine 20. In the absence of either SET8 or H4K20 residue, substantial genome-wide chromatin decompaction occurs allowing excessive loading of the origin recognition complex (ORC) in the daughter cells. ORC overloading stimulates aberrant recruitment of the MCM2-7 complex that promotes single-stranded DNA formation and DNA damage. Restoring chromatin compaction restrains excess replication licensing and loss of genome integrity. Our findings identify a cell cycle-specific mechanism whereby fine-tuned chromatin relaxation suppresses excessive detrimental replication licensing and maintains genome integrity at the cellular transition from mitosis to G1 phase.

[1] Biotech Research and Innovation Centre (BRIC), Faculty of Health and Medical Sciences, University of Copenhagen, Ole Maaløes Vej 5, 2200 Copenhagen N, Denmark. [2] Centre for Gene Regulation & Expression, School of Life Sciences, University of Dundee, Dow Street, Dundee DD1 5EH, UK. [3] Institut de Recherche en Cancérologie de Montpellier (IRCM), INSERM U1194, University of Montpellier, Institut Régional du Cancer (ICM), F-34298 Montpellier, France. [4] Institute for Molecular Biology and Medicine, Universite Libré de Bruxelles, Charleroi 6041, Belgium. [5] Institut de Génétique Moléculaire de Montpellier, University of Montpellier, CNRS, Montpellier 34293, France. [6] Centre National de la Recherche Scientifique (CNRS), Montpellier 34000, France. Correspondence and requests for materials should be addressed to C.S.S. (email: claus.storgaard@bric.ku.dk)

In eukaryotic cells, dynamic changes in chromatin structure and compaction are essential for proper progression through different stages of cell cycle and the maintenance of genome integrity[1]. During mitosis and cell division, chromatin is packaged into highly condensed mitotic chromosomes that promote error-free segregation of genetic material. Upon mitotic exit, chromosomes must rapidly switch from compact to more relaxed interphase structures that facilitate all DNA-based processes, by allowing access to enzymatic machineries involved in transcription and DNA replication or repair. It is widely believed that changes in histone posttranslational modifications (PTMs) largely contribute to regulate cell cycle chromatin organization by creating local and pan-nuclear (global) chromatin higher-order structures, which in turn define nuclear functions[2–4].

Histone phosphorylation and acetylation have been shown to correlate with compact and open chromatin structures, respectively, during cell cycle transitions. In particular, phosphorylation on histone H3 serine 10 and 28 and threonine 3, 6, and 11 increase significantly during the passage from relaxed interphase chromatin structures to condensed mitotic chromosomes[5–7]. Histone acetylation, on the other hand, creates a less compact chromatin structure by disrupting electrostatic interactions between histones and DNA[2]. However, most of what is known about the role of histone PTMs in chromatin structural transitions over the cell cycle has come through research on the progression from interphase into mitosis. The precise role of histone PTMs in regulating the transition from compact mitotic chromosomes to decondensed interphase chromatin structures during M/G1 transition is currently unresolved.

At the exit of mitosis, the transition from highly compact chromatin to a less compact interphase chromatin overlaps with the loading of replication origin licensing factors, in particular the ORC complex, which are essential for executing proper DNA replication[8]. ORC serves as a scaffold for the subsequent association of CDC6 and CDT1, which together coordinate the loading of the MCM2-7 complex in order to form the pre-replication complex (pre-RC) required for replication fork formation and activity. In metazoans, the absence of sequence specificity for ORC binding to DNA indicates that the local chromatin environment, defined by nucleosome positioning and histone modifications, might influence ORC recruitment to promote proper licensing of replication origins[9,10]. Whether chromatin compaction changes that occur from M to G1 phase impact ORC chromatin association and the establishment of replication origins remains unknown.

SET8, the mono-methyltransferase for histone H4 lysine 20 methylation (H4K20me) has previously been shown to be important for cell cycle progression and maintenance of genome integrity[11–14]. SET8 and H4K20me peak during G2 and M phases of the cell cycle, and this prompted us to investigate their involvement in chromatin compaction upon mitotic exit. Intriguingly, we find that SET8 and H4K20me are crucial for maintaining a chromatin compaction threshold during the cellular transition from mitosis to G1 phase, which suppresses aberrant DNA replication licensing. Furthermore, we show that loss of genome stability follows aberrant replication licensing. Together, our results uncover a key cell cycle-specific mechanism whereby chromatin structure limits DNA replication licensing and promote genome integrity throughout the cellular transition from M to G1 phase.

## Results

### SET8 maintains chromatin compaction in cells exiting mitosis.

We hypothesized that SET8 could regulate chromatin structure when cells transit from mitosis (M) to G1 phase. To test this, we first compared the chromatin compaction status of cells arrested in M with those in G1 in the presence or absence of SET8 using micrococcal nuclease (MNase) digestion assay. To avoid the deleterious impact of long-term SET8 depletion, we depleted the enzyme for maximally 21 h before harvesting cells (Fig. 1a–c). Cells were simultaneously labeled with methyl-[14]C containing thymidine during the experiment. After MNase digestion, methyl-[14]C released into the supernatant was used as a measure of compaction status of the cells (Methods). The more decompacted and accessible the chromatin is, the more methyl-[14]C is released into the supernatant. Notably, the compaction state of both control and siSET8 cells in mitosis were very similar (judged by the amount of methyl-[14]C released into the supernatant) (Fig. 1d). In contrast, SET8-depleted cells displayed a higher level of methyl-[14]C compared to control cells upon progression into G1 phase. This data suggests that SET8 likely contributes to maintain ground-state chromatin compaction in cells exiting mitosis.

To complement the results obtained from MNase assay, we investigated the genome-wide landscape of chromatin accessibility in G1 phase after SET8 depletion. We employed high-throughput sequencing-based assay of transposase accessible chromatin (ATAC-seq)[15]. To this end, we synchronized and small interfering RNA (siRNA) transfected cells as described in Fig. 1a (without nocodazole block) followed by harvesting cells in the following G1 phase. Supplementary Fig. 1a show the average distribution of ATAC-seq peaks in siSET8 vs siControl samples. Importantly, when visualizing the global signal intensity (Supplementary Fig. 1b) and signal normalized to the number of reads at individual loci (Supplementary Fig. 1c), it was evident that signal strength was higher in siSET8 cells. These data are consistent with the overall loss of chromatin compaction in the absence of SET8 as also observed in the MNase assay (Fig. 1d).

To further explore this notion in single and live cells, we performed quantitative analysis of chromatin compaction at the scale of nucleosome arrays using a FLIM-FRET (fluorescence lifetime imaging microscopy-Förster resonance energy transfer) approach in synchronized cells co-expressing histones H2B-EGFP and mCherry-H2B (named U2OS$_{H2B-2FPs}$). FRET was measured between fluorescent protein-tagged histones on separate nucleosomes, where an increase in signal signifies chromatin compaction[16]. siRNA-treated confluent cells were diluted in the presence of thymidine to synchronize them at the G1/S transition, and FRET signals were detected and spatially analyzed before and after release from the thymidine block. siControl and siSET8 cells showed similar compaction profiles as judged by the FRET efficiency map at the time of release from thymidine (T0) (Fig. 1e, f and Supplementary Fig. 2a–c). In contrast, we observed a significant reduction in FRET levels in siSET8 G1 phase cells, indicating a major reduction in the levels of chromatin compaction of these cells compared to control cells (Fig. 1e, f). To further confirm that SET8 regulates chromatin compaction status in cells exiting mitosis, we performed a similar FRET-based analysis and compared the chromatin compaction in cells arrested in G2/M vs G1 cells (Supplementary Fig. 3a, b). In agreement with our MNase digestion analysis (Fig. 1b), we detected a significantly lower mean FRET efficiency in siSET8 cells in G1 phase, but not at G2/M phases, compared to siControl cells (Supplementary Fig. 3c, d). Consistent with these results, transmission electron microscopy (TEM) analysis of siControl and siSET8 cells also revealed a reduction in chromatin density throughout the nucleus in SET8-depleted cells in G1 phase (Fig. 1g, h). Altogether these results indicate a major role for SET8 in securing appropriate chromatin compaction during the cellular transition from mitosis to G1 phase of the cell cycle.

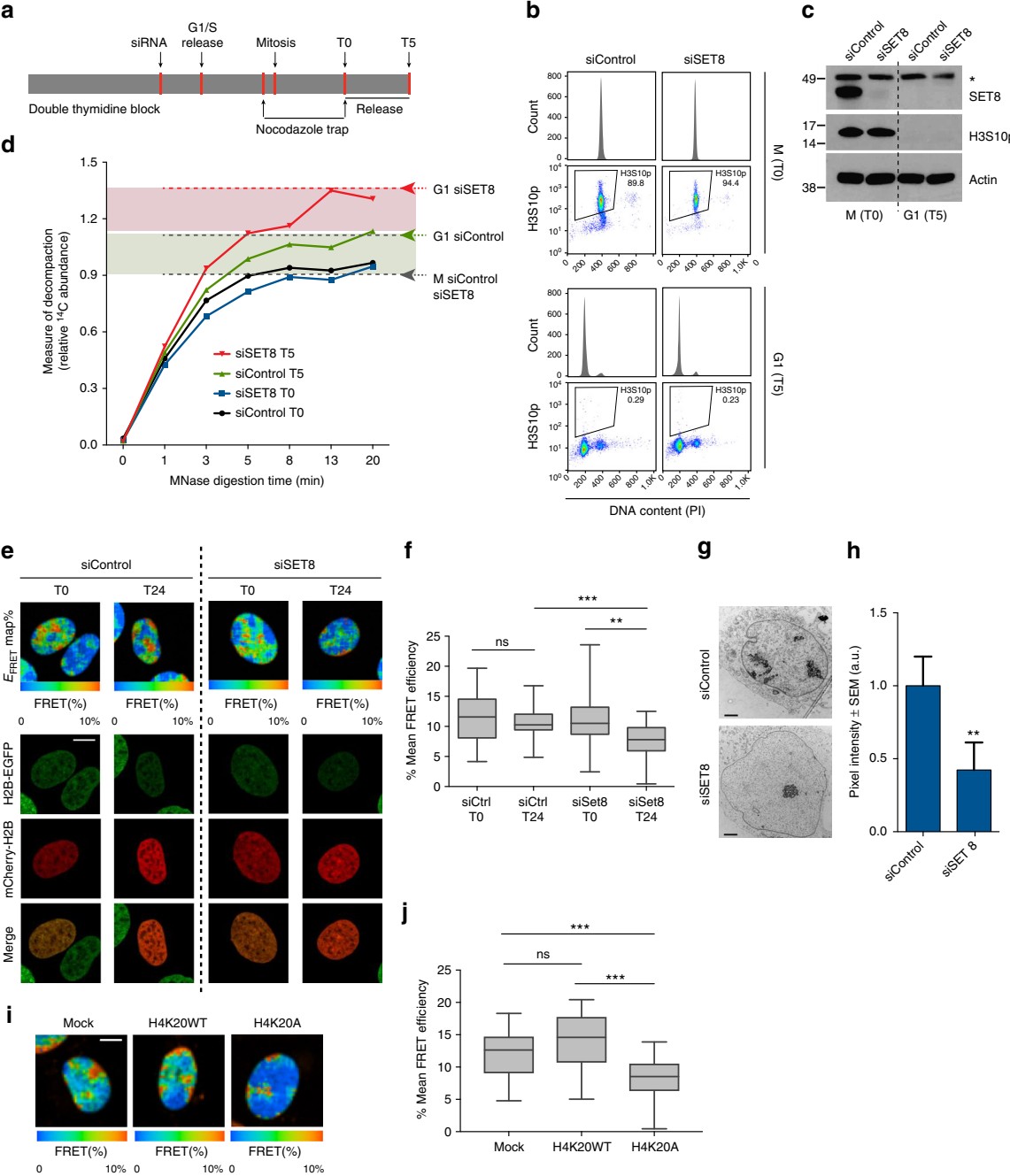

**Fig. 1** SET8 and H4K20 methylation regulate ground-state chromatin compaction in cells exiting mitosis. **a** U2OS cells were synchronized with double thymidine block. Control and SET8 siRNAs transfected 6 h before G1/S release. Cells were then blocked in mitosis with nocodazole for 4 h (T0) and released into G1 phase for 5 h (T5). MNase digestion was performed on mitotic arrested and G1 phase cells. Thymidine containing methyl-[14]C was added throughout the experiment. **b** Cells from **a** were fixed and stained with phospho-Histone H3S10 antibody and propidium iodide (PI) followed by flow cytometric analysis. **c** Immunoblots of total cell lysates prepared from the samples in **a** probed with the indicated antibodies. Asterisk (*) represents a non-specific band. **d** Graph showing MNase digestion profile. Levels of methyl-[14]C in the supernatant indicates the degree of chromatin decompaction over time when incubated with MNase. **e** U2OS cells stably expressing H2B-GFP alone (U2OS$_{H2B-GFP}$) or with mCherry-tagged histone H2B (U2OS$_{H2B-2FPs}$) were synchronized with single thymidine block. Cells were treated with either control or SET8 siRNA during the block. FRET measurements were taken before (T0) and 24 h after release (Bar, 10 μm). **f** Quantification of the FLIM-FRET chromatin compaction assay. Inside box plots, the thick line represents median, the boxes correspond to the mean FRET values upper or lower of the median, with whiskers extending to the highest and lowest value. $n > 30$ nuclei, ***$p < 0.001$, **$p < 0.01$ (ANOVA), ns not significant. **g** U2OS cells were synchronized and siRNA transfected as in **a**. Cells were fixed at 15 h post release for transmission electron microscope (TEM) visualization. **h** Quantification of the average pixel intensity ± SD ($n > 15$) of nuclei in **g**. **$p < 0.01$ (unpaired $t$ test). **i** U2OS cells stably expressing H2B-GFP alone (U2OS$_{H2B-GFP}$) or with mCherry-H2B (U2OS$_{H2B-2FPs}$) were transduced with FLAG-tagged Histone H4WT or H4K20A mutant. Mock transduced cells were taken as control. FRET measurements were taken for all the samples (Bar, 10 μm). **j** Quantification of the mean FRET levels in Mock, H4K20WT-, and H4K20A-expressing cells. Box plots represents mean FRET values as defined earlier (**f**). $n > 30$ nuclei. ns not significant, ***$p < 0.001$ (ANOVA)

**SET8 compacts chromatin through histone H4 lysine 20 methylation**. SET8 is responsible for the methylation of histone H4 at lysine 20, which has previously been implicated in chromatin compaction in in vitro assays[17]. Furthermore, H4 tail interaction with an acidic patch on H2A/H2B histones on neighboring nucleosomes has also been suggested to be important for maintaining ground state chromatin structure[18]. We therefore set out to investigate whether SET8 regulates chromatin compaction state through H4K20 methylation in cells. To achieve this, we again used the FLIM-FRET approach described earlier and transduced U2OS$_{H2B-2FPs}$ cells with a high titer of retroviral vectors encoding a FLAG-tagged histone H4 mutant carrying a lysine 20 to alanine substitution (H4K20A). Cells transduced with a virus encoding a FLAG-tagged wild-type histone H4 (H4K20WT) and mock-transduced cells were used as controls. After 3 days of viral transduction, FRET efficiency was detected and spatially analyzed. Immunoblot analysis revealed that FLAG-tagged H4K20WT and H4K20A proteins were expressed at similar levels and were efficiently incorporated into chromatin leading to a marked decrease in the global levels of the mono-methylated H4K20 (Supplementary Fig. 3e). FRET maps revealed a significant decrease in the FRET levels in cells expressing the H4K20A mutant version of histone H4 as compared to mock and H4K20WT-expressing cells (Fig. 1i, j). A similar decrease in mean FRET percentage was observed in the case of histone H4 lysine 20 to arginine (H4K20R) mutant-expressing cells (Supplementary Fig. 3f and g). Altogether, these data strongly suggest that SET8 maintains ground-state chromatin compaction via histone H4K20 methylation.

**Ground-state chromatin compaction in G1 phase promotes genome stability**. SET8 has previously been shown to be critical for safeguarding genome stability as evident from the appearance of DNA damage, cell cycle defects, and early embryonic lethality in SET8 knockout mice[11–13,19]. Since we observed a notable decrease in chromatin compaction in cells exiting mitosis in the absence of SET8, we investigated whether loss of genome stability parallels compaction status at this stage of cell cycle. To this end, cells were synchronized with a double thymidine block, treated with SET8 or control siRNA, for 6 h during the second block, and then released from the G1/S transition before analysis for cell cycle progression and the presence of DNA damage (Fig. 2a). Our data revealed that cells lacking SET8 and H4K20me1 go through the first S phase without DNA damage and only display DNA damage upon mitotic exit. This DNA damage accumulated as siSET8 cells approach S-phase entry, as evidenced by flow cytometric profiles of γH2A.X-positive cells (Fig. 2b–d and supplementary Fig. 4a, b). In addition, we observed an elevated γH2A.X nuclear staining (Fig. 2e) and the presence of DNA double-strand breaks on pulsed field gel electrophoresis and neutral COMET assay in siSET8 cells harvested at 15 h from G1/S release (Fig. 2f, g and Supplementary Fig. 4c). To further investigate the relationship between chromatin compaction and genome stability, we analyzed, in a similar experimental set-up as for SET8, the genome integrity in the presence of a histone deacetylase inhibitor (HDACi), which represents a well-known tool to induce genome-wide chromatin relaxation[20,21] (Fig. 2a). Short treatment of G1-phase synchronized cells with HDACi, i.e. Trichostatin A (TSA), induced DNA damage (Supplementary Fig. 4d–g) that is reminiscent of DNA damage observed in the absence of SET8 (Fig. 2b–e). Taken together, these results indicate that maintenance of chromatin compaction status during the cellular transition from M to G1 phase is critical for safeguarding genome integrity.

To verify that DNA damage upon loss of SET8 is not a consequence of improper mitotic progression, we analyzed synchronized U2OS cells arrested in metaphase (using

nocodazole) or released into G1 phase (Supplementary Fig. 5a). Our results showed that both siControl and siSET8 cells exit mitosis and enter G1 phase without notable delay and without any initial measurable DNA damage. siSET8 cells, however, progressively accumulated γH2A.X in the daughter cells (Supplementary Fig. 5b). The appearance of DNA damage in the daughter cells correlates well with the role of SET8 and H4K20me in maintaining ground-state chromatin compaction in cells exiting mitosis.

Next, we sought to understand whether SET8 maintains genome integrity through H4K20 methylation. To test this, we developed doxycycline (DOX)-inducible cell lines expressing either FLAG-HA-tagged wild-type histone H4 (H4K20WT) or FLAG-HA-tagged histone H4 mutant carrying a lysine to alanine or arginine substitution at position 20 (H4K20A/R) (supplementary Fig. 5c) We double thymidine blocked these cells and released them into the cell cycle using our standard protocol. Cells were fixed and stained for γH2A.X that revealed increased DNA damage signaling in both H4K20A- and H4K20R-expressing cells as compared to H4K20WT-expressing cells (Fig. 2h, i).

**Forced chromatin compaction rescues genome integrity after SET8 depletion**. To further understand the relationship between DNA damage accumulation and chromatin structure, we asked whether the siSET8 phenotype could be rescued by inducing global chromatin compaction. To achieve this, we used sucrose, which has been shown to induce molecular crowding and promote highly reversible chromatin compaction[22,23]. Consistently, in a similar experimental set-up previously described (Fig. 2a), the addition of sucrose in late mitosis induced a more compact chromatin state as cells reached G1 phase (Supplementary Fig. 6a, b). Second, we ectopically expressed RNF2, a component of the PRC1 complex, which can compact chromatin independent of its ubiquitin ligase activity[24]. As expected, TEM micrographs showed that RNF2 expression induced more compact chromatin in siControl and siSET8 cells (Supplementary Fig. 6a, b). To study the effects of chromatin re-compaction on genome stability, we used a similar experimental set-up as described earlier (Fig. 2a). In agreement with our hypothesis, addition of sucrose effectively suppressed DNA damage in cells lacking SET8 (Fig. 2j and Supplementary Fig. 6c). Similar to sucrose treatment, ectopic expression of RNF2 suppressed the challenge to genome integrity in siSET8 cells (Fig. 2k and Supplementary Fig. 6d, e). Taken together, these results suggest that maintenance of genome integrity in cells exiting mitosis and progressing through G1 phase depends on the degree of chromatin compaction set by the SET8-H4K20me pathway.

**Chromatin compaction threshold restricts excessive loading of licensing proteins**. During late mitosis and early G1 phase of the cell cycle, the six-subunit origin recognition complex (ORC), together with CDC6 and CDT1, loads the replicative helicase complex MCM2-7 onto DNA, a process also termed replication licensing or pre-RC formation[25–27]. As SET8 and H4K20me have also emerged as regulators of replication origin licensing[28,29], we wondered whether the ability of SET8 to ensure chromatin compaction in cells exiting mitosis could impact licensing. To address this question, we first examined the levels of ORC1 and MCM2 proteins selected as licensing markers. Cells were pre-extracted to remove soluble proteins prior to fixation and antibody staining procedure. This approach revealed increased nuclear abundance of pre-RC proteins in siSET8 cells (Fig. 3a–c). Similarly, an increase in the chromatin loading of replication proteins was observed in siSET8 cells in G1 phase by immunoblot analysis (Fig. 3d, e and Supplementary Fig. 7a).

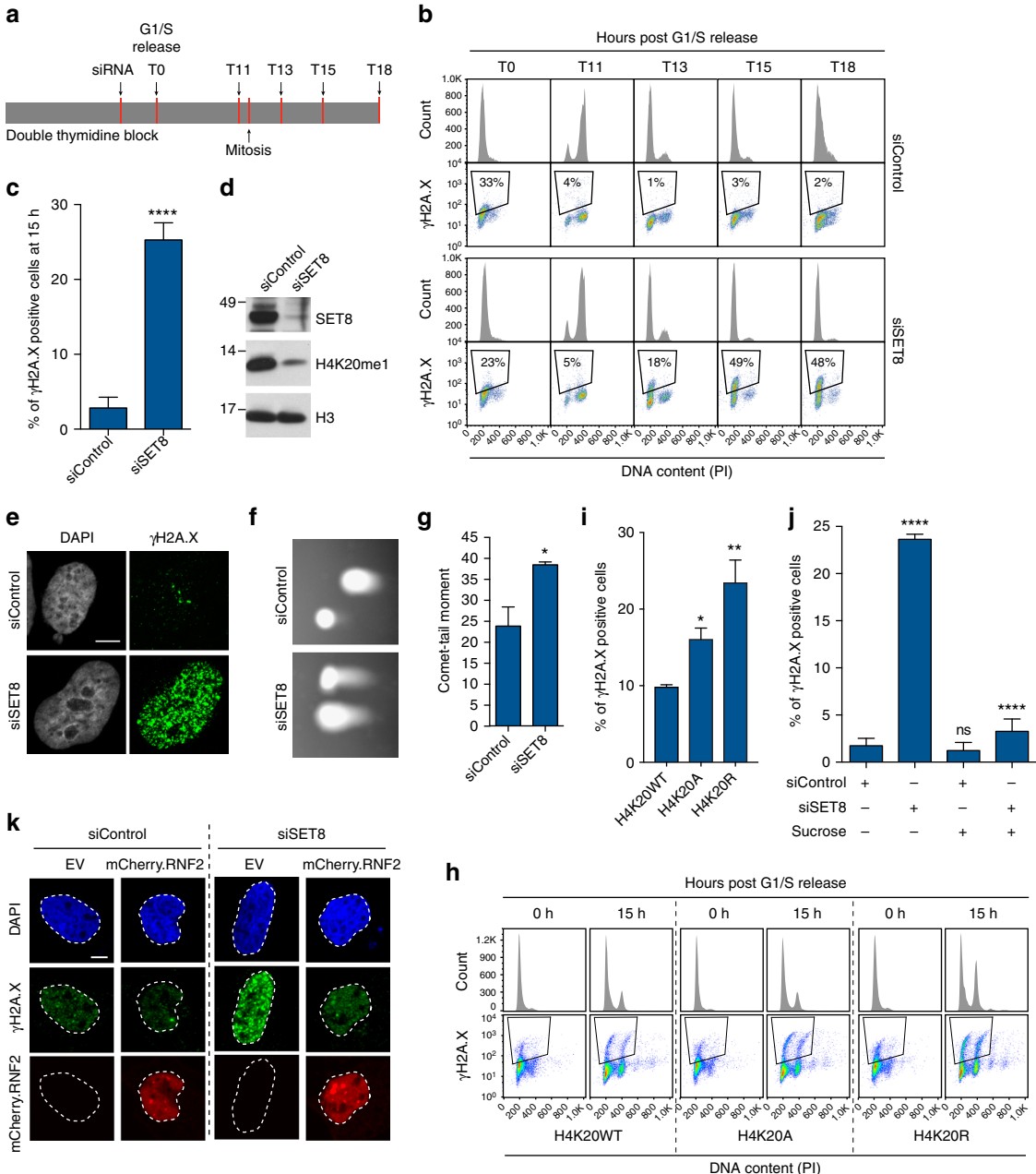

**Fig. 2** Maintenance of ground-state chromatin compaction ensures genome integrity **a** Design of the experiment. U2OS cells synchronized by double thymidine block were transfected with Control and SET8 siRNA. Cells were released into thymidine-free medium and harvested at the indicated time points. **b** Cells from **a** were fixed and stained with γH2A.X antibody and PI followed by flow cytometric analysis. **c** Bars represent percentage of γH2A.X-positive cells harvested at 15 h post G1/S release. Average ± SD of five independent experiments. ****$p < 0.0001$ (unpaired $t$ test). $n > 20,000$ in each experiment. **d** U2OS cells synchronized as in **a** were harvested at 15 h post G1/S release and immunoblotted with the indicated antibodies. **e** U2OS cells synchronized and siRNA transfected as in **a** were fixed at 15 h post G1/S release and stained with DAPI and γH2A.X antibody. (Bar, 10 μm). **f** U2OS cells synchronized and siRNA transfected as in **a** were prepared for single-cell electrophoresis in neutral electrophoresis buffer at 15 h post G1/S release. **g** Relative comet-tail moments from experiments in **f** plotted as means ± S.E.M. *$p < 0.05$ (unpaired $t$ test). $n > 55$. **h** U2OS cells expressing Dox-inducible FLAG-HA-tagged histone H4-wild type (H4K20WT) or histone H4 lysine 20 to alanine or arginine (H4K20A/R) mutant were synchronized with double thymidine block. Cells were fixed at 15 h post G1/S release and stained with γH2A.X antibody and PI followed by flow cytometric analysis. **i** Bars represent percentage of γH2A.X-positive cells from h. Average ± SD of three independent experiments. *$p < 0.05$, **$p < 0.01$ (ANOVA). $n > 20,000$ in each experiment. **j** U2OS cells synchronized and siRNA transfected as in **a** were mock and sucrose treated (125 mM) at 12 h post G1/S release and were fixed at 15 h post G1/S release. Bars represent percentage of γH2A.X-positive cells in the indicated samples (average ± SD of three independent experiments). ****$p < 0.0001$ (unpaired $t$ test). $n > 20,000$ in each experiment. **k** U2OS cells were transfected with mCherry-tagged human RNF2 (RING1b) and were double thymidine synchronized, siRNA transfected and fixed as in (2e) and stained with DAPI and γH2A.X antibody. (Bar, 10 μm)

To test whether this increase in chromatin loading of pre-RC proteins is related to H4K20me, we employed DOX-inducible H4K20WT- or H4K20A-expressing cell lines. Cells were synchronized with double thymidine block as previously described (Fig. 2a). To induce the expression of histone H4K20WT/A variants, DOX was added at the start of the experiment. Cells were harvested at 15 h post G1/S release and analyzed for levels of the ORC1 and MCM2 licensing markers. Our results revealed significantly higher levels of chromatin-bound ORC1 and MCM2 in cells expressing H4K20A as compared to H4K20WT-expressing cells (Fig. 3f–i). These results strongly support the notion that SET8-mediated H4K20 methylation creates a chromatin environment that limits the amount of ORC and MCM recruited on chromatin in G1 phase. Accordingly, we also noticed an increase in ORC1 and MCM2 staining in cells treated with an HDACi (TSA) in a similar experimental set-up (Supplementary Fig. 7b–e), suggesting that overloading of these pre-RC components is caused by alterations in the levels of chromatin compaction. Furthermore, MCM loading was significantly restricted when cells were treated with sucrose-based hypertonic medium to induce a more compact chromatin environment (Fig. 3e and Supplementary Fig. 7f, g). Altogether, these results support our notion that chromatin compaction regulates the replication origin licensing process, likely by limiting the accessibility of DNA that may serve as pre-RC-binding sites.

**Chromatin structure limits accumulation of ssDNA**. Next, we investigated how aberrant replication licensing can impact genome integrity. In this regard, we first investigated the phosphorylation of MCM2 on serine 53 (MCM2-S53p) by Cdc7/Dbf4-dependent kinase (DDK), which is thought to be an essential step in the activation of the replicative helicase, starting at G1/S border[30,31]. Interestingly, we found that the MCM2-S53p is markedly increased in siSET8 cells (Supplementary Fig. 8a, b). A consequence of MCM activation is DNA unwinding leading to the formation of single-stranded DNA (ssDNA), which is a key step in the replication process. Such unwinding is normally strictly regulated spatially and temporally, occurring only at a fraction of replication origins starting at the G1/S boundary and continuing throughout S phase[32]. We asked whether alteration in chromatin compaction status could lead to accumulation of ssDNA after brief depletion of SET8. For this purpose, we analyzed native bromodeoxyuridine (BrdU) staining as ssDNA readout in synchronized cells progressing toward the G1/S transition. We pulse labeled cells with BrdU at the time of release from double thymidine block and fixed them in next cell cycle (15 h from G1/S release). Notably, native BrdU signal was highly abundant in cells lacking SET8, suggesting the presence of ssDNA as compared to that in the control situation (Fig. 4a, b). Consistently, the major ssDNA-binding protein RPA[33,34] was increased on chromatin in the absence of SET8 further implying the presence of ssDNA (Fig. 4c and Supplementary Fig. 8c, d).

To further investigate the role of chromatin structure in controlling replication licensing and preventing DNA unwinding, we analyzed both native BrdU signal and RPA chromatin loading in cells treated with sucrose, in the same experimental set-up as in Fig. 4a. Indeed, native BrdU signal and RPA loading were significantly reduced by adding hypertonic medium to cells lacking SET8 (Fig. 4a–c and Supplementary Fig. 8c). Taken together, these data indicate that chromatin compaction threshold prevents DNA unwinding possibly by limiting the chromatin association of pre-RC components in cells progressing through G1 phase.

**Chromatin-mediated suppression of the MCM2-7 complex promotes genome integrity**. Our results suggested an important role for suppression of MCM activity in chromatin compaction-dependent genome integrity. To further test this, we first depleted the MCM7 subunit of the MCM2-7 complex to levels sufficient to still allow cell cycle progression (Supplementary Fig. 9a, b). Conspicuously, reducing MCM7 protein levels inhibited the challenge to genome integrity in cells lacking SET8 (Fig. 4d, e). Importantly, ssDNA levels were also reduced after MCM7 co-depletion in cells lacking SET8 (Supplementary Fig. 9b).

DDK-dependent phosphorylation is required for activation of MCM2-7 helicase activity[35,36]. Moreover, in yeast, it has been reported that an MCM5 mutant mimics CDC7-dependent MCM-complex phosphorylation and activates the helicase activity of the MCM-complex leading to aberrant DNA unwinding[37]. Therefore, to further verify the functional involvement of MCMs in cells with perturbed chromatin compaction, we determined whether DDK activity contributes to ssDNA accumulation and γH2A.X signaling. Notably, we observed a dramatic reduction in DNA damage, as evident from γH2A.X-positive cells, when two different DDK inhibitors (PHA-767491 and XL413)[38–40] were added to the cells lacking SET8 (Fig. 4f and Supplementary Fig. 9c). Moreover, treatment of siSET8 cells with DDK inhibitor reduced ssDNA accumulation (Supplementary Fig. 9d). These results support our hypothesis that chromatin structure plays an important role in proper loading and timing of activation of licensing factors.

Finally, we sought to determine whether chromatin relaxation precedes DNA damage rather than the alternative scenario where DNA damage leads to chromatin relaxation. To this end, we performed MNase digestion of nuclei lacking SET8 and simultaneously treated with DDKi to suppress MCM-dependent genome instability. The results revealed that siSET8 cells retained their relaxed chromatin even in the absence of DNA damage (Supplementary Fig. 9e). Collectively, these results suggest that abnormal chromatin relaxation precedes events that lead to the loss of genome integrity.

**Discussion**
Here we identify a tightly regulated chromatin compaction threshold, whereby SET8-mediated H4K20 methylation limits replication licensing (Fig. 4g, h). This ensures proper replication to maintain genome stability through the cell cycle. These results provide a novel link between cell cycle-specific chromatin structure regulation and genome integrity.

Our data regarding the role of H4K20me in maintaining ground-state chromatin compaction is in agreement with previous in vitro studies, in which histone H4 tail domains have been shown to induce short-range nucleosome–nucleosome interactions contributing to local array compaction and higher-order chromatin folding[41,42]. Notably, nucleosome crystallization studies revealed that histone H4 tail residues from lysine 16 to iso-leucine 26 interacts with an H2A/H2B acidic patch on a neighboring nucleosome[18,43,44]. In vitro studies also revealed that, in addition to acidic patch interaction, a region of the H4 tail close to the histone fold domain mediates internucleosomal interactions through direct contacts to both DNA and protein targets in condensed chromatin structures[45,46]. In this regard, H4K20me may favor more stable H4 internucleosomal interactions either through increased H4 tail–acidic patch interactions or via H4 tail–DNA interactions or both, which is consistent with our results obtained in single-cell-based FRET assay. Further, we find that global chromatin compaction in mitosis is not affected in cells lacking SET8 and proper H4K20me levels. The high degree of condensation during mitosis may be more dependent on other factors, such as the SMC complex proteins[47–50].

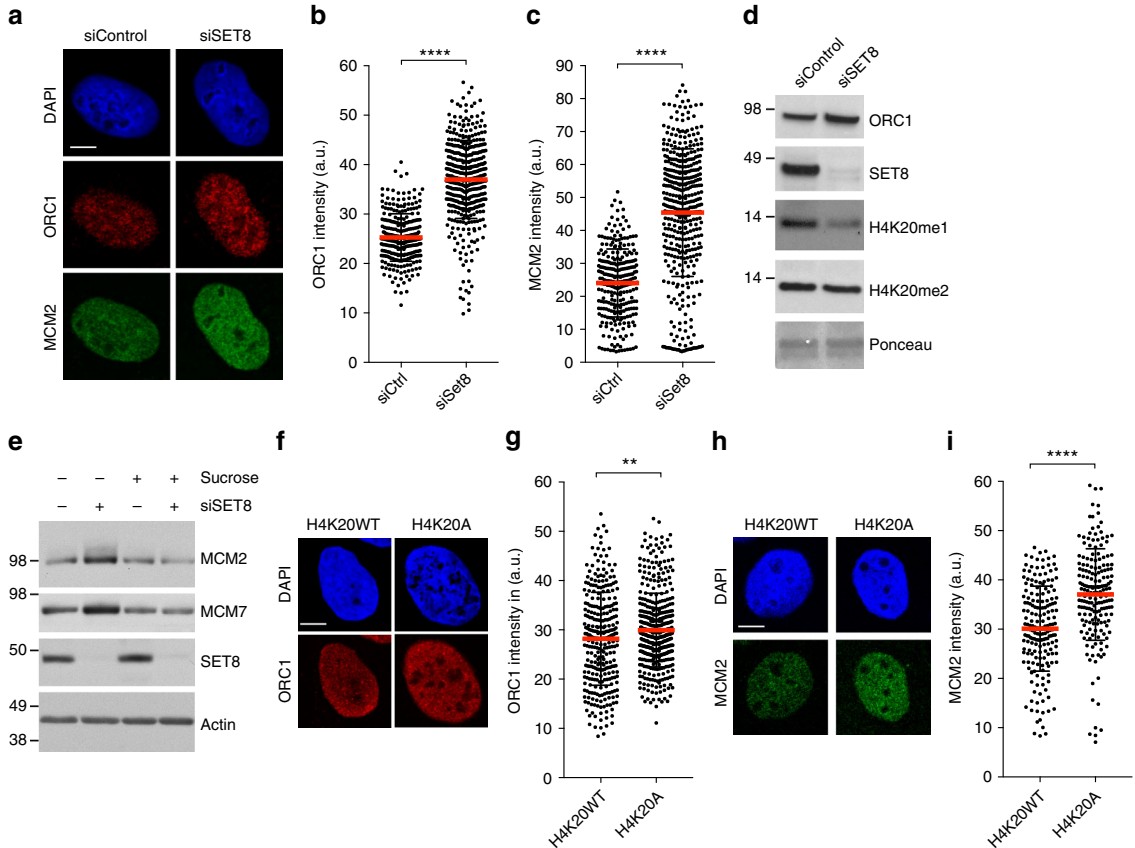

**Fig. 3** Chromatin compaction threshold restricts excessive loading of replication licensing factors. **a** U2OS cells were synchronized with double thymidine block, transfected with either siControl or siSET8 6 h before G1/S release, and fixed at 15 h post release. Cells were pre-extracted in CSK buffer containing 0.5% triton and immunostained with the indicated antibodies. Cells were counterstained with DAPI (Bar, 10 μm). **b** Scatter plot showing the quantification of ORC1 intensity from cells in **a** where mean ± SD is indicated by red lines. $n > 150$, $****p < 0.0001$ (unpaired $t$ test). $n > 280$. **c** Scatter plot showing the quantification of MCM2 intensity from cells in **a** where mean ± SD is indicated by red lines. $n > 150$, $****p < 0.0001$ (unpaired $t$ test). $n > 280$. **d** Chromatin fraction was prepared from synchronized U2OS cells depleted as in **a**, harvested at 15 h, and immunoblotted with the indicated antibodies. **e** Chromatin was prepared from cells synchronized and transfected with siRNAs as in **a**, treated with sucrose at 12 h post G1/S release, and harvested at 15 h. Samples were blotted with the indicated antibodies. **f** U2OS cells expressing DOX-inducible FLAG-HA-tagged histone H4WT or H4K20A mutant were synchronized as in **a** and fixed at 15 h post release. Cells were pre-extracted and immunostained with the indicated antibody as well as DAPI for DNA (Bar, 10 μm). **g** Scatter plot showing the quantification of ORC1 intensity from HA-positive cells in **f** where mean ± SD is indicated by red lines. $n > 250$, $**p < 0.01$ (unpaired $t$ test). **h** U2OS cells expressing DOX-inducible FLAG-HA-tagged histone H4WT or H4K20A mutant were synchronized as in **a** and fixed at 15 h post release. Cells were pre-extracted and immunostained with the indicated antibody as well as DAPI for DNA (Bar, 10 μm). **i** Scatter plot showing the quantification of MCM2 intensity from HA-positive cells in **h** where mean ± SD is indicated by red lines. $n > 180$, $****p < 0.0001$ (unpaired $t$ test)

Our data suggest that a histone H4K20me-dependent chromatin compaction threshold imposes constraints and hence regulates the chromatin loading of ORC and the MCM2-7 complexes. This is consistent with recent reports using in vitro replication assays to suggest that chromatin enforces origin specificity by suppressing non-specific ORC binding[51,52]. The licensing of replication origins can, therefore, be viewed as an opportunistic mechanism aided by the ORC complex's (and MCM2-7 complex's) affinity for DNA per se, where ORC bound at future replication origins may promote MCM2-7 complex loading and replication initiation by establishing a permissive nucleosome-free chromatin environment[52,53]. Consistently, the affinity of ORC1 and ORC-associated protein (ORCA) to H4K20me marks and the role of SET8 in the maintenance of properly compact chromatin structure would contribute to create a restricted number of high ORC affinity sites at specific positions along the genome[28,29,54]. Therefore, in G1 cells depleted for SET8, our data suggest that both the impairment of SET8-mediated high affinity for ORC proteins and the de-compacted chromatin environment lead to opportunistic binding of ORC/MCM complexes to DNA, thereby causing this promiscuous overloading in G1 cell cycle phase. Furthermore, increased loading of licensing factors in the context of decompacted chromatin may not only allow for increased availability of the substrate to the activating kinases (DDK/CDKs) but also facilitates access for these S phase kinases, thereby promoting accumulation of ssDNA. Thus we suggest that moderately compacted chromatin with appropriate H4K20 methylation levels limits the number of potentially available ORC-binding sites for the formation of replication origins and in turn keep a check on activating kinases.

Previous studies linked SET8 and H4K20me with a positive role in licensing[29,54]. However, these studies were carried out under extended periods of analysis (e.g., >72 h in Tardat et al.[29]), when loss of SET8 activity significantly reduced H4K20me2 and H4K20me3 levels and thus affected the stability of ORC complex binding to chromatin[28,29]. Furthermore, these studies mainly focused on the consequence of SET8 stabilization during S phase. In the current study, we used relatively short-term depletion of

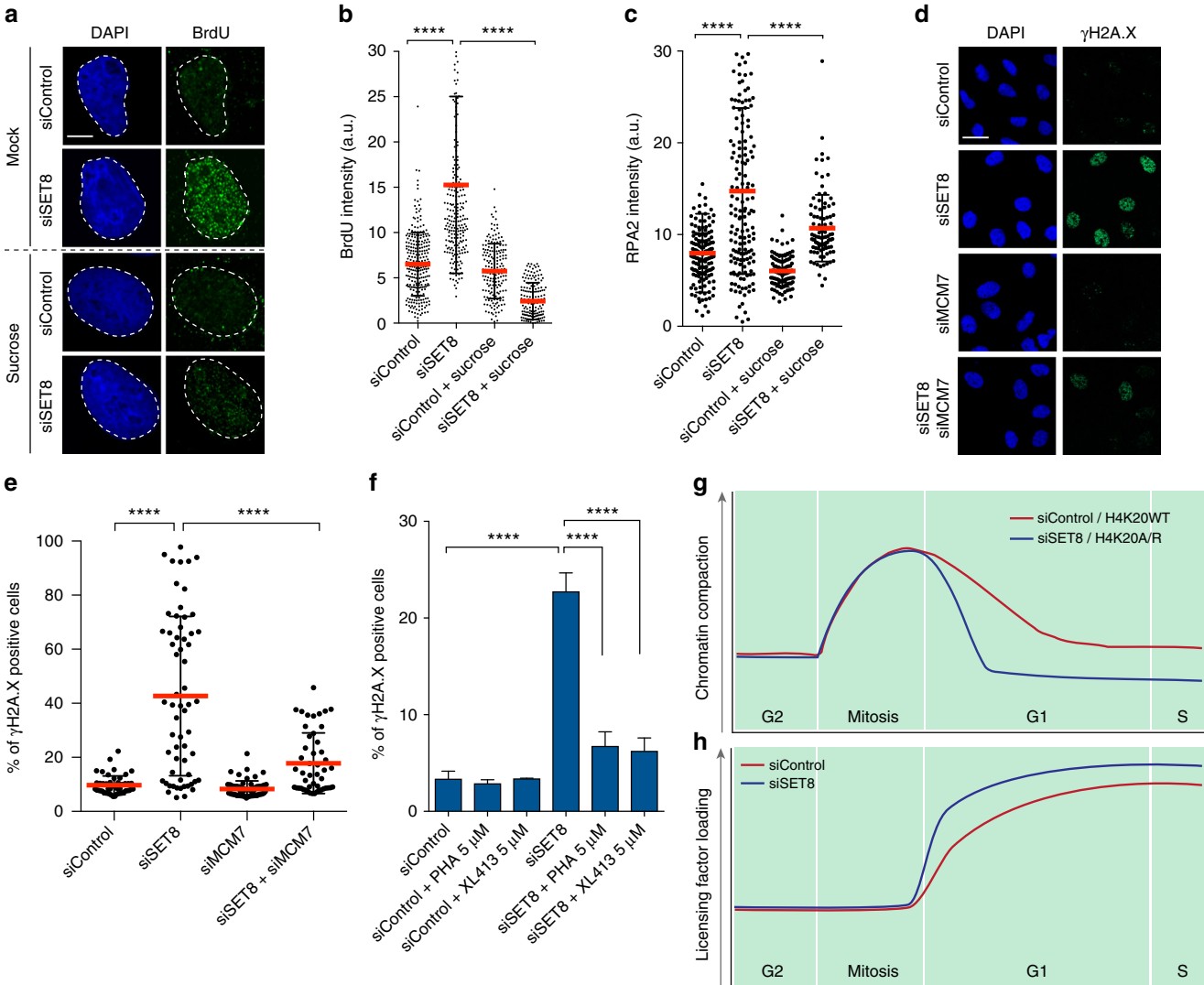

**Fig. 4** Chromatin-mediated suppression of the MCM2-7 complex promotes genome integrity. **a** U2OS cells synchronized by double thymidine block were transfected with Control and SET8 siRNA. Cells were released into BrdU-containing medium from G1/S boundary and were mock or sucrose treated at 12 h post G1/S release. Cells were then fixed at 15 h post G1/S release and were immunostained with DAPI and γH2A.X antibody (Bar, 10 μm). **b** Scatter plot showing the quantification of BrdU intensity (**a**). Mean ± SD is indicated by red lines. n > 230, ****p < 0.0001 (ANOVA). **c** U2OS cells were synchronized, siRNA transfected, treated with sucrose, and fixed as in **a**. Samples were immunostained with an RPA2 antibody after pre-extraction. Scatter plot showing the quantification of RPA2 intensity. Mean ± SD is indicated by red lines. n > 150, ****p < 0.0001 (ANOVA). **d** U2OS cells were transfected with siMCM7 and synchronized with double thymidine block. SET8 was depleted 6 h before G1/S release. Cells were fixed at 15 h post release and processed for immunofluorescence staining with DAPI and γH2A.X antibody (Bar, 30 μm). **e** Scatter plot showing the quantification of γH2A.X intensity from cells in **d**. Mean ± SD is indicated by red lines. n > 50, ****p < 0.0001 (ANOVA). **f** U2OS cells were synchronized and transfected as in **a**. DDKi inhibitors (PHA-767491 and XL413) were added at 11 h post release from G1/S boundary. Cells were then collected at 15 h post release and processed for FACS staining with PI and γH2A.X antibody. Bars representing the percentage of γH2A.X-positive cells in the indicated samples (average ± SD of three independent experiments). ****p < 0.0001 (unpaired t test). n > 20,000 in each experiment. **g** Illustrative plot details the dynamics of chromatin compaction/decompaction over different phases of cell cycle. Loss of SET8 and H4K20me leads to a notable reduction in chromatin compaction status in cells exiting mitosis. **h** Illustrative plot details the dynamics of loading of replication licensing factors. In late mitosis, the licensing starts with loading of ORC complex that promotes loading of MCM2-7 complex throughout G1 phase. Loss of SET8 and H4K20me favors excessive loading of ORC and MCM2-7 complexes in daughter cells

SET8 (21 h) in synchronized cells, thereby allowing the persistence of H4K20me2 (Fig. 3d) that serves as an ORC recruiting chromatin mark. Notably, this approach allowed us to uncover specifically the role of SET8-mediated chromatin compaction in replication licensing after a single passage through mitosis. Of note, our finding that general HDAC inhibition shows a highly similar effect, which is mediated by chromatin structure control, further supports a general role of chromatin compaction in the regulation of replication licensing process.

SET8 has previously been shown to promote genome stability and proper S-phase progression[11,12]. In the current work, we demonstrated that appearance of DNA damage in the absence of SET8 parallels with the loss of ground-state chromatin compaction and aberrant licensing in cells exiting mitosis. These events are followed by abnormal activation of MCM helicase and an accumulation of ssDNA. Thus our findings shed light on a fundamental role of SET8 in maintaining chromatin structure, thereby explaining initial events leading to appearance of high

levels of replication stress and DNA damage during S phase in cells lacking SET8[11,12]. Furthermore, our data indicate that SET8 regulates chromatin compaction and maintains genome stability specifically via histone H4K20 methylation. Overall, these findings support the notion that chromatin structural organization in G1 phase allows for fine-tuned regulation of DNA-based processes, such as replication, thereby preventing replication stress and endogenous damage[55].

## Methods

**Cell culture, cell cycle synchronization, and chemicals**. U2OS were obtained from ATCC and maintained in Dulbecco's modified Eagle's medium containing 10% fetal bovine serum and 1% Penicillin–Streptomycin. For synchronization at G1/S, U2OS cells were cultured in the presence of 2 mM thymidine (Sigma) for 20 h, washed three times with phosphate-buffered saline (PBS), and released in fresh medium without thymidine for 10 h. After another 17 h in thymidine, cells were washed three times with PBS and cultured in fresh medium. Cells were then collected at 15 h. DOX-inducible stable U2OS cell lines were generated using Lenti-X Tet-One-inducible expression system (Clontech Laboratories). Briefly, N-terminal FLAG-HA-tagged histone H4 as HindIII-EcoRI fragment was PCR amplified from pcDNA4/TO-FLAG-HA-H4 and cloned into pLVX-TetOne-Puro vector to generate pLVX-H4WT using the In-Fusion-HD Cloning Kit (Clontech Laboratories). pLVX-H4K20A/R variants were generated using site-directed mutagenesis. For constitutively expressing histone H4WT and H4K20A/R variants, pQCXIP-H4WT was generated by cloning the histone H4-3xFLAG into pQCXIP (retroviral vector, Clonetech) by PCR using pCMV-H4-3xFLAG vector. pQCXIP H4K20A substitution mutant was generated by site-directed mutagenesis[56]. The list of primers is provided as supplementary table 1 in supplementary information. For various cellular treatments, the following drugs were used: 3 µM DDKi (PHA-767491 and XL413 were from Sigma), TSA 25 µM (Sigma), sucrose 125 mM (Sigma), nocodazole 40 ng per ml (Sigma), DOX 1 µg per ml (Sigma).

**siRNA transfections**. siRNA transfections were performed with 20 nM siRNA duplexes using Lipofectamine® RNAiMAX (Invitrogen), according to the manufacturer's instructions. The siRNA sequences used for knockdown are (5'–3'):
　　SET8 (GUACGGAGCGCCAUGAAGU) and
　　MCM7 (UAGCCUACCUCUACAAUGA).

**Flow cytometry**. Cells were fixed in 70% ethanol and stained with the indicated antibodies for 1 h followed by 1 h incubation with the secondary antibodies. DNA was stained using 0.1 mg/ml propidium iodide containing RNase for 30 min at 37 °C. Flow cytometric analysis was performed on FACSCalibur using the CellQuest Pro software (BD). Data were analyzed using the FlowJo software (v7.2.2; Tree Star). The details of all primary and secondary antibodies used in this study are provided as Supplementary Tables 2 and 5.

**Cellular fractionation and chromatin isolation**. To obtain soluble and chromatin-enriched cellular fractions, cell fractionation was performed. In brief, cells were lysed for 10 min in a small volume of CSK buffer (0.5% Triton X-100, 10 mM Pipes, pH 6.8, 300 mM sucrose, 100 mM NaCl, and 1.5 mM MgCl₂). The lysed cells were pelleted by centrifugation at $2000 \times g$, and the supernatant was collected (soluble fraction). The pellet was washed once with CSK buffer, resuspended in 0.2 M HCl, and incubated at 4 °C for 2 h. The supernatant represented the chromatin-enriched fraction. HCl-containing samples were neutralized with Tris buffer before sodium dodecyl sulfate–polyacrylamide gel electrophoresis (SDS–PAGE). To obtain cytoplasmic, nuclear, and chromatin fractions, cellular fractionation was performed as previously described[29,57]. Briefly, cells were resuspended in buffer A (10 mM HEPES, [pH 7.9], 10 mM KCl, 1.5 mM MgCl₂, 0.34 M sucrose, 10% glycerol, 1 mM dithiothreitol (DTT), 5 µg of aprotinin per ml, 5 µg/ml leupeptin, 0.5 µg/ml pepstatin A, 0.1 mM phenylmethylsulfonyl fluoride (PMSF)). Triton X-100 (0.1%) was added, and the cells were incubated for 5 min on ice. Nuclei were collected in pellet 1 (P1) by low-speed centrifugation (4 min, $1300 \times g$, 4 °C). The supernatant (S1) was further clarified by high-speed centrifugation (15 min, $20,000 \times g$, 4 °C) to remove cell debris and insoluble aggregates (S2). Nuclei were washed once in buffer A and then lysed in buffer B (3 mM EDTA, 0.2 mM EGTA, 1 mM DTT, protease inhibitors as described above). Insoluble chromatin (S3) was collected by centrifugation (4 min, $1700 \times g$, 4 °C), washed once in buffer B, and centrifuged again under the same conditions. The final chromatin pellet (P3) was resuspended in Laemmli buffer and sonicated.

**Immunoblotting and antibodies**. Cells were lysed on ice in cold EBC-buffer (150 mM NaCl; 50 mM TRIS pH 7.4; 1 mM EDTA; 0.5% NP-40/Igepal) containing protease inhibitors (1% aprotinin, 5 µg/ml leupeptin, 1 mM PMSF), phosphatase inhibitors (50 mM sodium fluoride; β-glycerophosphate; 0.5 µM Calyculin A) and 1 mM DTT. The lysates were sonicated using a digital sonifier (102C CE Converter; Branson). Proteins were separated by SDS–PAGE and transferred to a nitrocellulose membrane. Blocking and blotting with primary antibodies were performed in PBS-T supplemented with 5% skimmed milk powder. Proteins were visualized on films using secondary horseradish peroxidase-conjugated antibodies and ECL (GE Healthcare). Films were developed using an X-ray machine (Valsoe; Ferrania Technologies). The details of all primary and secondary antibodies used in this study are provided as Supplementary Tables 3 and 5. Supplementary Fig. 10 contains uncropped scans of the immunoblots.

**Immunofluorescence microscopy**. Cells were grown on coverslips, washed with PBS, fixed with formaldehyde 4% for 10 min, permeabilized with PBS containing 0.3% Triton X-100 for 10 min at room temperature (RT) and blocked for 1 h in PBS containing 0.1% Triton X-100 and 3% bovine serum albumin prior to incubation with the indicated antibodies. For visualizing chromatin-bound proteins, cells grown on coverslips were washed with PBS, then extracted on ice for 3 min in ice-cold pre-extraction buffer (0.5% Triton X-100, 20 mM HEPES, pH 7.5, 300 mM sucrose, 50 mM NaCl, and 3 mM MgCl₂), washed twice with PBS, and fixed with 4% formaldehyde for 10 min at RT and then incubated in primary antibodies followed by fluorochrome-labeled secondary antibodies. DAPI (4',6-diamidino-2-phenylindole, dihydrochloride) was used to counter stain the nuclei. Images were acquired using either Leica TCS SP8 confocal microscope or Zeiss LSM880 in AiryScan super-resolution mode. The details of all primary and secondary antibodies used in this study are provided as Supplementary Tables 4 and 5.

**Micrococcal nuclease digestion**. Two million U2OS cells were labeled with ¹⁴C (radioactive isotope of Carbon) during synchronization with double thymidine and were harvested in G1 phase. Nuclei were prepared as described previously[58]. Briefly, cells for each condition were resuspended in cytosolic lysis buffer (10 mM Tris-HCl, pH 7.5, 10 mM NaCl, 5 mM MgCl₂, 0.5% NP-40, and 0.25 mM PMSF) and incubated on ice for 8 min. Nuclei were pelleted by centrifugation ($1700 \times g$ for 10 min at 4 °C). The pellet was washed once in nuclei buffer (60 mM KCl, 15 mM NaCl, 0.34 M sucrose, 0.25 mM PMSF, and 1 mM DDT) and resuspended in nuclei buffer. CaCl₂ (2 mM) was added, and the samples were pre-warmed to 25 °C. Micrococcal nuclease (0.1 U/µl; Sigma-Aldrich) was added to each sample and aliquoted into 7 pre-chilled eppendorf tubes. Six tubes were incubated at 37 °C for the indicated time periods (0, 1, 3, 8, 13, and 20 min) and 1 tube from each sample was sonicated as a control for ¹⁴C incorporation efficiency in the cells. All samples were immediately centrifuged at $10,000 \times g$ and supernatants were collected in the scintillation tubes containing 4 ml of scintillation liquid (Ultima Gold, Perkin Elmer). All samples were quantified using scintillation counter.

**Electron microscopy**. Cells were fixed in Karnofski solution (3% paraformaldehyde, 0.5% glutaraldehyde in 10 mM PBS, pH 7.4) for 1 h, washed once in PBS, and post-fixed first in 1% reduced osmium tetroxide (containing 1.5% potassium ferricyanide) for 40 min and subsequently in 1% osmium tetroxide for another 40 min. After washing in water, fixed samples were dehydrated in an ascending ethanol series, embedded in Epon resin (Fluka, Buchs, Switzerland). Thin sections were cut on a Reichert Ultracut microtome (Reichert-Jung Optische Werke, Vienna, Austria) using a diamond knife (Diatome, Biel, Switzerland). The sections were collected on parlodion-coated copper grids and stained with 6% uranylacetate for 1 h followed by 2% lead citrate for 2 min[59]. EM micrographs were recorded on a Phillips CM-100 transmission electron microscope equipped with a CCD camera at an acceleration voltage of 80 kV. Images were recorded using the systems software and processed using Adobe Photoshop. Quantification of chromatin density was performed using ImageJ. Briefly, on each image 15 points/areas were randomly selected and pixel density was measured. Pixel density of the background was measured at five random image points outside the cell. Background density was subtracted from the measured chromatin density and values were normalized relative to chromatin density in control siRNA-treated cells set as 1 (Fig. 1h). To avoid negative relative values, values in Supplementary Fig. 6b were normalized to the lowest value (siSET8), which was defined as 1.

**FLIM-FRET measurements and analysis**. FLIM-FRET experiments were performed in U2OS cells stably expressing H2B-GFP alone (U2OS_H2B-GFP) or with mCherry-tagged histone H2B (U2OS_H2B-2FPs). FLIM was performed using an inverted laser scanning multiphoton microscope LSM780 (Zeiss) equipped with temperature- and CO₂-controlled environmental black wall chamber. Measurements were acquired in live cells at 37 °C, 5% CO₂, and with a ×40 oil immersion lens NA 1.3 Plan-Apochromat objective from Zeiss. Two-photon excitation was achieved using a Chameleon Ultra II tunable (680–1080 nm) laser (Coherent) to pump a mode-locked frequency-doubled Ti:Sapphire laser that provided sub-150-femtosecond pulses at a 80 Mhz repetition rate with an output power of 3.3 W at the peak of the tuning curve (800 nm). Enhanced detection of the emitted photons was afforded by the use of the HPM-100 module (Hamamatsu R10467-40 GaAsP hybrid PMT tube). The fluorescence lifetime imaging capability was provided by TCSPC electronics (SPC-830; Becker & Hickl GmbH). TCSPC measures the time elapsed between laser pulses and the fluorescence photons. Enhanced green fluorescent protein (EGFP) and mCherry fluorophores were used as a FRET pair. The optimal two-photon excitation wavelength to excite the donor (EGFP) was

890 nm. Laser power was adjusted to give a mean photon count rate of the order $1 \times 10^5 - 5 \times 10^5$ photons/s. Fluorescence lifetime measurements were acquired over 60 s and fluorescence lifetimes were calculated for all pixels in the field of view (256 × 256 pixel). The analysis of the FLIM measurements was performed by using the SPCImage software (Becker & Hickl, GmbH). Because FRET interactions cause a decrease in the fluorescence lifetime of the donor molecules (EGFP), the FRET efficiency was calculated by comparing the FLIM values obtained for the EGFP donor fluorophores in the presence (U2OS$_{H2B-2FPs}$) and absence (U2OS$_{H2B-GFP}$) of the mCherry acceptor fluorophores. FRET efficiency (E FRET) was derived by applying the following equation:

$$E\,FRET = 1 - (\tau DA/\tau D)$$

where $\tau DA$ is the mean fluorescence lifetime of the donor (H2B-EGFP) in the presence of the acceptor mCherry-H2B in U2OS$_{H2B-2FPs}$ cells and $\tau D$ is the mean fluorescence lifetime of H2B-EGFP (in the absence of acceptor) in U2OSH2BGFP cells that are present in the same field of view. FRET efficiency values were calculated from 20 to 30 cells and then normalized. Graphical representation was done using the GraphPad Prism software.

**Assay of transposase accessible chromatin sequencing.** ATAC-seq was performed as originally described by Buenrostro et al.[15]. Briefly, nuclei were prepared by spinning 50,000 cells at $500 \times g$ for 5 min, followed by washing with ice-cold 1× PBS followed by centrifugation at $500 \times g$ for 5 min. Cells were lysed using cold lysis buffer (10 mM Tris-HCl, pH 7.4, 10 mM NaCl, 3 mM MgCl$_2$, and 0.1% NP40, followed by centrifugation at $500 \times g$ for 10 min using a refrigerated centrifuge. Following the nuclei preparations, the pellet was resuspended in the transposase reaction mix (25 μl 2× TD buffer, 2.5 μl transposase (Illumina), and 22.5 μl nuclease-free water) and incubated at 37 °C for 30 min. The sample was purified using a Qiagen MinElute Kit. After purification, the DNA fragments were amplified using Nextera PCR master mix (NPM) and 1.25 μM of custom Nextera (Illumina) PCR primers 1 and 2, using the following PCR conditions: 72 °C for 5 min; 98 °C for 30 s; and thermocycling at 98 °C for 10 s, 63 °C for 30 s, and 72 °C for 1 min. We performed the size selection (<600 bp) using Ampure XP magnetic beads (Beckman Coulter Inc.) according to manufacturer's protocol. To reduce GC and size bias in our PCR, we performed a quantitative real-time PCR (qPCR)-based library quantification. First, one fifth of the purified PCR product was amplified using 2x KAPA SYBR FAST qPCR Master mix (KK4932) for 40 cycles. The optimal number of cycles were determined by the cycle number that corresponds to one third of maximum fluorescent intensity (usually around 7–8 cycles). The full libraries were then amplified for the corresponding number of cycles (determined in previous step) for each sample. The libraries were again then purified with size selection (<600 bp) using Ampure XP magnetic beads according to the manufacturer's protocol. Libraries were quantified using the Qubit DNA HS Kit, and for quality control, 1 μl of each sample was run on Bioanalyzer High Sensitivity DNA Chip. In all, 4 nM of all libraries were pooled and 1.5 pM were analyzed on Illumina NextSeq500 (500/550 High Output v2 Kit—150 cycles).

The raw paired-end reads were first trimmed for Nextera transposase adapter sequences using Trimmomatic (v0.32) in palindrome mode with default settings except ILLUMINACLIP:2:30:10:1:true MINLEN:25. FastQC of reads before and after trimming confirmed the removal of any 3' adapter sequences, while also clearly showing the known insertion Tn5 motif in the 5'-ends. The trimmed PE reads were mapped to the hg19 assembly (canonical chromosomes only) using bowtie2 v.2.2.9 with default settings except -k 2 -X 2000 --no-mixed --no-discordant. After sorting (SortSam) and labeling duplicates (MarkDuplicates) with Picard tools (v. 2.6.0-27-g915ffa7-SNAPSHOT) and adding a NH tag (number of reported alignments), reads were filtered to exclude unmapped, multimapping, and mitochondrial reads (samtools view -f 2 -F 4 and custom filter). The filtered bam files were converted to bed format using bedtools bamtobed (v2.26.0-92-g88cd6c5), and read start and stop coordinates were finally adjusted by +5 bp and −4 bp, respectively, to adjust for Tn5-binding properties as previously described[15].

ATAC-seq peaks were identified individually for each set of data using macs2 (v2.1.1.20160309)[60] callpeak broad -f BAMPE -t $f -g hs -q 0.05, intersected using bedtools[61] multiinter -I, and regions positive in at least two sets were merged within 1 kbp of each other using bedtools merge -i 1000. Subsequent handling and visualization was done using EaSeq (v1.05)[62]. Values in scatter plots were quantified within a 1-kbp window surrounding the center of each region using the Quantify-tool, quantile normalized using the Normalize-tool, and averaged for all replicates in Microsoft Excel. Tracks were visualized using the FillTrack-tool and replicates were made transparent and superimposed in Adobe Illustrator.

## Data availability
The ATAC-seq data presented in this article is deposited in GEO database: GSE118606.

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

## Acknowledgements

We are grateful to S. Jørgensen and C. Colding for assistance in the early stages of the project and to Yasuko Antoku and Kasper Nørskov Kragh for immunofluorescence analysis assistance. We thank Ursula Sauder and Vesna Oliveri for electron microscopy expert technical assistance. We thank Brian Larsen for his help in Comet assay and Heike Ilona Rösner for useful scientific discussions and advice. We also thank Sung-Bao Lee for help in setting up MNase assay. We thank Montpellier Ressources Imagerie (MRI) for assistance with FLIM-FRET microscopy. C.S.S., M.S., and D.W. are funded by The Novo Nordisk Foundation (to C.S.S.), The Danish Cancer Society (to C.S.S.), The Lundbeck Foundation (to C.S.S.), and The Danish Medical Research Council (to C.S.S.), Swiss National Science Foundation (to D.W.), and Villum Foundation (to M.S.). B.F. is funded by FNRS Belgium and P.J.G. and J.J.B. are funded by Cancer Research UK (grant C303/A14301). F.I. and E.J. were supported by grants from French Plan-Cancer (EPIG2013-13), Labex EpiGenMed, and SIRIC Montpellier Cancer. F.I. was supported by a PhD fellowship from the French Ligue Contre le Cancer and Fondation pour la Recherche Médicale. D.L. was supported by a Cancéropole GSO-Emergence grant (2014-E17) and CNRS.

## Author contributions

M.S., D.W., and P.J.G. designed, performed, and analyzed the experiments. F.I and D.L. generated the H4K20 mutants in 2FP-expressing U2OS cells and performed and analyzed the FLIM-FRET experiments. M.L. and J.V.J. analyzed the ATAC-seq data. B.F. performed electron microscopic analysis. C.S.S., K.H., E.J., and J.J.B. directed the project. C.S.S. and M.S. conceived and designed the project. M.S., C.S.S., and E.J. wrote the manuscript.

## Additional information

**Competing interests:** The authors declare no competing interests.

