## [Peer Review File · Nature Communications]

Reviewers' Comments:

Reviewer #2:

Remarks to the Author:

The revised paper by Shoaib et al. explores how the lysine methyltransferase Set8 regulates chromatin compaction in the mitosis to G1 transition of the cell cycle. The major claim is that Set8 maintains a level of chromatin compaction during the transition, and loss of Set8 during the M to G1 transition causes chromatin decompaction and subsequent DNA damage. The paper has strong quantitative experiments with multiple complementary experiments to change chromatin compaction. The ability to manipulate histone H4 by overproducing a mutant is striking, as well as RNF2 based chromatin compaction. The authors present strong evidence that Set8 depletion leads to less chromatin compaction in G1 phase and that this change is correlated with increased endogenous DNA damage. It is clear that Set8 depletion or histone H4K20 mutation leads to more ORC and MCM binding at the time points tested, and the short time scale preserving H4K20me2 explains why these results are not contradicting the previous work of Tardat et al 2010.

What isn't yet convincing is that the change in ORC and MCM binding is the true cause of the DNA damage markers. Certainly these two molecular events are correlated, but there are many other potential explanations for DNA damage from a global change in chromatin compaction, and earlier origin licensing could be just one of many symptoms of that global change. I certainly think that this dataset is valuable to the field and should be made available, but the presentation should be more cautious about suggesting that the DNA damage is principally caused by the preRC effects. There are so many possible causes of DNA damage given the global chromatin changes shown.

Major points:

1. The authors should test for DNA damage markers in S and G2 phase of the preceding cell cycle in Set8-depleted cells. It is possible that the G1 DNA damage was acquired in the previous cell cycle. The absence of strong γ -H2AX signal in the mitotic-arrested cells isn't enough to rule this out because the DDR is known to be dampened in mitosis relative to interphase cells; for example Giunta et al, JCB 2010, PMID:20660628, but there are others.

2. The authors should determine if the γ -H2AX positive cells after mitotic release are truly in G1 and not a subset of cells that have already started S phase, but then arrested. Alternatively, they should revise their interpretations.

If the DNA damage is replication-dependent (which is almost surely the case), then the siSet8 cells could be entering S phase sooner than the single time point analyzed (T15), acquiring damage, and then arresting. Comparing T15 between siControl and siSet8 would then be comparing different cell cycle phases. This is supported by Figure R7 that the cells may have already entered S phase, arrested due to damage, and then stopped incorporating EdU by T15 (R7 should be moved to supplemental data). The suppression of damage from siSet8 by sucrose and RNF2, DDK inhibitors and even siMCM7 could just be because the cells treatments delay S phase entry or cause a G1 arrest. (Additionally, the siMCM7 experiment was so poorly synchronized that one can't tell what phase the cells are in.)

The same concern applies to measures of MCM activation by phosphorylation, RPA loading, ssDNA detection, etc, all of these are markers of S phase. If the manipulations lead to a higher fraction of cells entering S phase at the single time point typically analyzed, then it isn't really "unscheduled sDNA formation." As written, the use of "premature unwinding" suggests that the cells are definitely still in G1 phase with active MCM, and that's not confirmed. Similarly "excessive DNA unwinding" which implies a higher fraction of active origins or runaway helicase isn't confirmed.

This major concern can be addressed simply by adjusting the text to avoid making a causal connection that isn't fully supported. The rest of the paper is very solid, convincing, and superbly presented. If they authors are determined to claim activation of MCM within G1 by some

mechanism other than normal S phase origin firing (which by definition isn't G1 anymore), then they should examine multiple earlier timepoints than T15 or biochemical markers of S phase. This addition would determine if the siSet8 cells are crossing the G1/S boundary faster than siControl. Moreover do the manipulations that rescue DNA damage change cell cycle progression or cause a G1 arrest?

Minor points.

- Please indicate the number of cells analyzed for each experiment in the figure legends
- For immunofluorescence quantifications, are the measurements total nuclear signal or average signal per unit area? This measurement matters because the nuclear area clearly changes with chromatin compaction.
- Figure legend for 1 is mislabeled b-d,
- Supplementary figure 3 is misidentified as Supp figure 1 on lines 126 and 127.
- which time point is represented in figures 2d, 2e, 2f, and 2g? the legend refers to Figure 2a, but there are multiple G1 time points in a.
- Legend to 3g and 3i indicate p values with 4 asterisks but the figures themselves have 2
- The authors should quantify the % H2AX positive in Figure 2k as they did for their other % H2AX as in Figure 4e.

Reviewer #3:

Remarks to the Author:

Shoib et al report that as dividing cells reach G1, a certain level of chromatin compaction is required to ensure normal replication origin licensing. This level is controlled through the action of histone methyltransferase Set8/PR-Set7 on H4K20, which favors chromatin compaction. Acute loss of Set8 leads to DNA damage, as assessed by gH2AX and detection of double strand breaks in neutral comet assays. This effect is linked to excessive origin licensing and premature activation of the MCM helicase, followed by an accumulation of ssDNA. The phenotypes caused by Set8 downregulation are alleviated by sucrose, a crowding agent that promotes reversible chromatin compaction, as well as by downregulation of MCM7 or inhibition of CDC7, the MCM-activating kinase.

This is a revised version of an earlier manuscript that I had reviewed for Nat Cell Biol in late 2015. Several of my comments and criticisms to the earlier version have been addressed. For instance, I had suggested that the authors should directly investigate the roles of Set8 and H4K20 methylation in chromatin compaction during G1, and this is now shown in a series of interesting experiments that include micrococcal nuclease chromatin digestion, ATAC-Seq, tagged-H2B FRET, and electron microscopy images (Figure 1 and Supp material). Therefore, the connection between Set8 and the changes in chromatin compaction at the time of origin licensing has been strengthened.

I had also asked the authors to investigate the timing of the generation of DNA breaks caused by loss of Set8. Several synchronization experiments are presented, including one with nocodazole block and release, that support the author's notion that most of the damage occurs in G1 phase.

In contrast, other aspects of the study remain less clear. My recommendation would be that the authors have the opportunity to further revise and clarify a few remaining issues before the paper can be accepted for publication.

Main comments:

1. The comet assay indeed points at DSBs, but how are they generated? The authors link them to the presumed activation of the MCM helicase and the generation of ssDNA stretches, based on native BrdU staining and RPA foci formation. Is the model that nucleases then act on exposed

ssDNA? Is this happening without actual DNA synthesis? This seems a key aspect of the model and could be easily tested by monitoring BrdU incorporation in G1.

2. As pointed out in the previous review, the conclusions of this study are very different from those of Tardat et al (Nat Cell Biol 2010), who reported that origin licensing coincides with an increase in H4 K20 methylation, and Set8 downregulation reduced the amount of chromatin-bound ORC, CDC6 and MCM proteins. The leading author in the earlier study is now a co-author in this article. Still, the discussion of the discrepancies between both studies is very vague. It is argued that previous studies "were carried out under extended periods of analysis (...) whereas we used relatively short-term depletion of SET8 (21h) in synchronized cells..." but the actual difference is not very large (21h in the current study vs 48h of shRNA in the earlier one, which also used synchronized cells -Fig 5A), and yet they yield completely opposite results. Furthermore, a compelling experiment in the earlier NCB paper showed that targeting Set8 to ectopic genomic sites resulted in the loading of ORC1, MCM2 and MCM5 (Fig 5C). It is hard to reconcile how Set8 both promotes and restricts pre-RC formation. This issue needs a more detailed discussion.

3. Another aspect that deserves discussion is whether the main effect of the reduced chromatin compaction caused by Set8 downregulation is the excessive loading of ORC MCM proteins, as apparently favored by the authors in the abstract, or the premature activation of Dbf4-Cdc7 kinase. As MCMs are already loaded in large excess relative to the number of active origins in WT cells, it is not clear that a modest increase in the total amount of MCMs would immediately lead to the reported increase in DNA damage. Is it not more intuitive that a context of reduced chromatin compaction would facilitate the access of activating kinases to licensed origins?

Minor points:

4. Please check and clarify Set8 immunoblot signal in Figure 1C (no Set8 in G1?)
5. Figure 1 legend does not follow the correct panel order, and the asterisk in 1c is not explained.
6. A more detailed explanation of the EM analysis of chromatin density should be provided (Fig 1G, H).
7. P 10, l 234, "Fig 3a" probably should read "Fig 2a".

Reviewers' comments:

Reviewer #2 (Remarks to the Author):

The revised paper by Shoaib et al. explores how the lysine methyltransferase Set8 regulates chromatin compaction in the mitosis to G1 transition of the cell cycle. The major claim is that Set8 maintains a level of chromatin compaction during the transition, and loss of Set8 during the M to G1 transition causes chromatin decompaction and subsequent DNA damage. The paper has strong quantitative experiments with multiple complementary experiments to change chromatin compaction. The ability to manipulate histone H4 by overproducing a mutant is striking, as well as RNF2 based chromatin compaction. The authors present strong evidence that Set8 depletion leads to less chromatin compaction in G1 phase and that this change is correlated with increased endogenous DNA damage. It is clear that Set8 depletion or histone H4K20 mutation leads to more ORC and MCM binding at the time points tested, and the short time scale preserving H4K20me2 explains why these results are not contradicting the previous work of Tardat et al 2010.

What isn't yet convincing is that the change in ORC and MCM binding is the true cause of the DNA damage markers. Certainly these two molecular events are correlated, but there are many other potential explanations for DNA damage from a global change in chromatin compaction, and earlier origin licensing could be just one of many symptoms of that global change. I certainly think that this dataset is valuable to the field and should be made available, but the presentation should be more cautious about suggesting that the DNA damage is principally caused by the preRC effects. There are so many possible causes of DNA damage given the global chromatin changes shown.

Authors response: We would like to thank the reviewer for their positive feedback on our manuscript. We understand and agree with the reviewer's concerns regarding the causal relationship between over licensing and DNA damage markers. In the revised manuscript we have adjusted the text as suggested by the reviewer.

Major points:

1. The authors should test for DNA damage markers in S and G2 phase of the preceding cell cycle in Set8-depleted cells. It is possible that the G1 DNA damage was acquired in the previous cell cycle. The absence of strong g-H2AX signal in the mitotic-arrested cells isn't enough to rule this out because the DDR is known to be dampened in mitosis relative to interphase cells; for example Giunta et al, JCB 2010, PMID:20660628, but there are others.

Authors response: We agree with the reviewer's concern that damage in second cell cycle could be a follow through from the previous cell cycle. However, our data showed that siSET8 treated cells when released from G1/S go through the first cell cycle without damage. As shown in Fig. 2b, at 11 hours post G1/S release, a time point where majority of cells are in G2 phase, the gH2AX levels are equivalent to background levels as in siControl treated cells.

2. The authors should determine if the g-H2AX positive cells after mitotic release are truly in G1 and not a subset of cells that have already started S phase, but then arrested. Alternatively, they should revise their interpretations.

If the DNA damage is replication-dependent (which is almost surely the case), then the siSet8 cells could be entering S phase sooner than the single time point analyzed (T15), acquiring damage, and then arresting. Comparing T15 between siControl and siSet8 would then be comparing different cell cycle phases. This is supported by Figure R7 that the cells may have already entered

S phase, arrested due to damage, and then stopped incorporating EdU by T15 (R7 should be moved to supplemental data). The suppression of damage from siSet8 by sucrose and RNF2, DDK inhibitors and even siMCM7 could just be because the cells treatments delay S phase entry or cause a G1 arrest. (Additionally, the siMCM7 experiment was so poorly synchronized that one can't tell what phase the cells are in.)

The same concern applies to measures of MCM activation by phosphorylation, RPA loading, ssDNA detection, etc, all of these are markers of S phase. If the manipulations lead to a higher fraction of cells entering S phase at the single time point typically analyzed, then it isn't really "unscheduled ssDNA formation." As written, the use of "premature unwinding" suggests that the cells are definitely still in G1 phase with active MCM, and that's not confirmed. Similarly "excessive DNA unwinding" which implies a higher fraction of active origins or runaway helicase isn't confirmed.

This major concern can be addressed simply by adjusting the text to avoid making a causal connection that isn't fully supported. The rest of the paper is very solid, convincing, and superbly presented. If they authors are determined to claim activation of MCM within G1 by some mechanism other than normal S phase origin firing (which by definition isn't G1 anymore), then they should examine multiple earlier timepoints than T15 or biochemical markers of S phase. This addition would determine if the siSet8 cells are crossing the G1/S boundary faster than siControl. Moreover do the manipulations that rescue DNA damage change cell cycle progression or cause a G1 arrest?

Authors response:

- We would like to thank the reviewer for highlighting these excellent points and for their appreciation of our work. We noticed that at 15hrs post G1/S release, 10-15% cells are entering S phase, as evident from detectable EdU incorporation in siControl cells (former Figure R7a,b). As suggested by the reviewer we have now moved Figure R7 to supplementary data (Supplementary Fig. 4a, b). We agree with the assumption that DNA damage in the siSET8 treated sample- arise in the population of cells that attempt to start S phase.
- We agree with the reviewer that the terms "unscheduled", "premature" and "excessive" ssDNA formation are not optimal as the appearance of ssDNA coincides with start of S phase. Wording in the manuscript has been modified to reflect this. We would like to point out that in the cells treated with siSET8, there is an accumulation of ssDNA as apparent from native BrdU and RPA staining (most likely at the onset of S phase). This large amount of RPA loading is not consistent with normal replication related loading of RPA.
- As far as the question whether this unwinding is due to higher fraction of active origins or due to a runaway helicase is concerned, we agree with the reviewer that our results do not address this. As per reviewer's suggestion, we have adjusted the manuscript accordingly.

Minor points.

- Please indicate the number of cells analyzed for each experiment in the figure legends

Authors response: We have added the number of cells for each experiment where quantification is shown.

- For immunofluorescence quantifications, are the measurements total nuclear signal or average

signal per unit area? This measurement matters because the nuclear area clearly changes with chromatin compaction.

Authors response: Although the IF quantifications represent average nuclear signal, we do see an increase in chromatin loading of licensing factors in biochemical fractions as shown in several WBs (Fig. 3d,e and Supplementary fig. 7a).

- Figure legend for 1 is mislabeled b-d,

Authors response: Corrected

- Supplementary figure 3 is misidentified as Supp figure 1 on lines 126 and 127.

Authors response: Corrected

- which time point is represented in figures 2d, 2e, 2f, and 2g? the legend refers to Figure 2a, but there are multiple G1 time points in a.

Authors response: Fig. 2d-e represents time point T15. This is now added in the legend.

- Legend to 3g and 3i indicate p values with 4 asterisks but the figures themselves have 2

Authors response: Corrected

- The authors should quantify the % H2AX positive in Figure 2k as they did for their other % H2AX as in Figure 4e.

Authors response: A new figure and graph are now added in the supplementary figures (Supplementary fig. 6e, f).

Reviewer #3 (Remarks to the Author):

Shoab et al report that as dividing cells reach G1, a certain level of chromatin compaction is required to ensure normal replication origin licensing. This level is controlled through the action of histone methyltransferase Set8/PR-Set7 on H4K20, which favors chromatin compaction. Acute loss of Set8 leads to DNA damage, as assessed by gH2AX and detection of double strand breaks in neutral comet assays. This effect is linked to excessive origin licensing and premature activation of the MCM helicase, followed by an accumulation of ssDNA. The phenotypes caused by Set8 downregulation are alleviated by sucrose, a crowding agent that promotes reversible chromatin compaction, as well as by downregulation of MCM7 or inhibition of CDC7, the MCM-activating kinase.

This is a revised version of an earlier manuscript that I had reviewed for Nat Cell Biol in late 2015. Several of my comments and criticisms to the earlier version have been addressed. For instance, I had suggested that the authors should directly investigate the roles of Set8 and H4K20 methylation in chromatin compaction during G1, and this is now shown in a series of interesting experiments that include micrococcal nuclease chromatin digestion, ATAC-Seq, tagged-H2B FRET, and electron microscopy images (Figure 1 and Supp material). Therefore, the connection between Set8 and the changes in chromatin compaction at the time of origin licensing has been strengthened.

I had also asked the authors to investigate the timing of the generation of DNA breaks caused by loss of Set8. Several synchronization experiments are presented, including one with nocodazole block and release, that support the author's notion that most of the damage occurs in G1 phase.

In contrast, other aspects of the study remain less clear. My recommendation would be that the authors have the opportunity to further revise and clarify a few remaining issues before the paper can be accepted for publication.

Authors response: We would like to thank the reviewer for the acknowledgment of our work and providing us with a positive feedback to further improve our manuscript.

Main comments:

1. The comet assay indeed points at DSBs, but how are they generated? The authors link them to the presumed activation of the MCM helicase and the generation of ssDNA stretches, based on native BrdU staining and RPA foci formation. Is the model that nucleases then act on exposed ssDNA? Is this happening without actual DNA synthesis? This seems a key aspect of the model and could be easily tested by monitoring BrdU incorporation in G1.

Authors response: The mechanism of generation of DNA breaks is unclear at the moment. The DNA damage arises at the onset of S phase (T15 where control cell start to replicate), there are two likely mechanisms based on our current insights;

- i) activation of MCM helicase leads to generation of ssDNA (here accompanied by deficiency in polymerase loading/activation) which is then targeted by nucleases,**
- ii) activated replication forks stall after initiation, which on one hand leads to uncoupling of DNA polymerase and MCM helicase creating long stretches of ssDNA, while on the other hand also leading to fork collapse. This generates DNA DSBs through nuclease processing that accompanies fork restart attempts.**

The EdU figure formerly rebuttal figure R7 is now in Supplementary data (Supplementary Fig. 4a, b) showing bulk replication is very much suppressed at the time of analysis. Thus, we favor scenario i) though this would have to be experimentally proven, which is a major project in itself.

2. As pointed out in the previous review, the conclusions of this study are very different from those of Tardat et al (Nat Cell Biol 2010), who reported that origin licensing coincides with an increase in H4 K20 methylation, and Set8 downregulation reduced the amount of chromatin-bound ORC, CDC6 and MCM proteins. The leading author in the earlier study is now a co-author in this article. Still, the discussion of the discrepancies between both studies is very vague. It is argued that previous studies “were carried out under extended periods of analysis (...) whereas we used relatively short-term depletion of SET8 (21h) in synchronized cells...” but the actual difference is not very large (21h in the current study vs 48h of shRNA in the earlier one, which also used synchronized cells -Fig 5A), and yet they yield completely opposite results. Furthermore, a compelling experiment in the earlier NCB paper showed that targeting Set8 to ectopic genomic sites resulted in the loading of ORC1, MCM2 and MCM5 (Fig 5C). It is hard to reconcile how Set8 both promotes and restricts pre-RC formation. This issue needs a more detailed discussion.

Authors response: We acknowledge that our data and previously published data on SET8’s role in replication licensing at a first glance appear to differ. However, we find that this can be readily explained by differences in experimental setup.

- Tardat et al depleted SET8 for more than 72 hours (48h shRNA+36h double-thymidine block), and the cellular fractionation protocol used by Tardat et al only investigated whether pre-RC components stably associate with chromatin at G1/S arrested cells. Our current study is focused on cells progressing dynamically from mitosis towards G1 phase after short SET8 depletion and on the measure of chromatin loading of ORC and MCM. It is therefore not surprising that we observe significant differences in experimental results when comparing 21h with 72h depletion. Furthermore, in contrast to our short-term analyses; loss of SET8 in Tardat et al strongly reduced all H4K20me states, which is expected to directly affect the stability of ORC complex binding to chromatin.
- Regarding the second part of reviewer’s concern, our data does not necessarily contradict SET8 dependent recruitment of licensing factors at ectopic loci. Localized ectopic SET8 will have a chromatin compacting function, however, such expression will also create a number of neighboring high affinity sites for ORC loading. The latter may well overrule the compacting impact of ectopic SET8.

A revised part related to these discussion points has been added in the discussion section (see page 15, lines 334-349).

3. Another aspect that deserves discussion is whether the main effect of the reduced chromatin compaction caused by Set8 downregulation is the excessive loading of ORC MCM proteins, as apparently favored by the authors in the abstract, or the premature activation of Dbf4-Cdc7 kinase. As MCMs are already loaded in large excess relative to the number of active origins in WT cells, it is not clear that a modest increase in the total amount of MCMs would immediately lead to the reported increase in DNA damage. Is it not more intuitive that a context of reduced chromatin compaction would facilitate the access of activating kinases to licensed origins?

Authors response: We thank the reviewer for pointing out this aspect. We agree with reviewer’s conclusion that reduced chromatin compaction may lead to increased accessibility of S phase

kinases to licensed origins. This is certainly an interesting area of research, which require further studies. We have now added comments on this aspect in the discussion section.

Minor points:

4. Please check and clarify Set8 immunoblot signal in Figure 1C (no Set8 in G1?)

Authors response: We believe that this difference is due to different synchronization strategies used e.g. in Fig 1c and Fig. 2b,d. (double thymidine block + Nocodazole shake off VS double thymidine block, respectively). Fig 1c represents cells released from Nocodazole shake off and collected 5 hrs later when they are in late G1 or perhaps very early S phase. Since they represent a relatively pure G1 population, SET8 is most likely very efficiently degraded by SCF β -TRCP in G1 phase. While in case of double thymidine block alone, after 15 hrs of G1/S release, there might be some residual SET8 from G2/M cells.

5. Figure 1 legend does not follow the correct panel order, and the asterisk in 1c is not explained.

Authors response: Corrected

6. A more detailed explanation of the EM analysis of chromatin density should be provided (Fig 1G, H).

Authors response: A detailed explanation of EM analysis is now added in the materials and methods section.

7. P 10, l 234, "Fig 3a" probably should read "Fig 2a".

Authors response: Corrected

Reviewer # 3 commenting on Reviewer #1 original concerns:

"This reviewer had raised as a general critique that the study fell short of demonstrating that chromatin compaction is directly required for genome stability through the loading and activation of helicase complexes in G1. Regarding this comment, which I shared, I believe that the authors have improved the study with new experiments that strongly support a role for Set8 in regulating chromatin compaction in G1.

Reviewer 1 also had a long list of specific points, and the authors have made an effort to address them. In most cases, the outcome is reasonable, e.g. (1) the insufficient resolution of some IF microscopy images has been corrected by using higher-resolution microscopy; (2) ectopic expression of RNF2 has been tried as an alternative to promote chromatin compaction; (3) the abundance of micronuclei has been quantified; (4) comet assays have been performed to monitor DNA damage; (5) nuclear area measurements with DAPI have been replaced by an alternative method based on FRET; (6) immunoblot cross-reacting species are explained.

Authors response: We thank the reviewer for their acknowledgement of our work.

In my opinion, one of the reviewers's comments (specific point 2) would deserve further clarification: upon Set8 downregulation, there is a risk of indirect effects caused by defective binding of 53BP1 protein to chromatin. The authors report that 53BP1 downregulation does not increase the percentage of gH2AX-positive cells, with or without codepletion of Set8 (Figure R1). The results, however, are slightly confusing, as not all si53BP1 molecules have similar effects, and two of them (#2 and 3) actually lead to less DNA damage following siSET8. It would be desirable if the authors could check directly the reviewer's concern by testing whether siSET8 affects the amount of 53BP1 on chromatin using biochemical fractionations, as done in other parts of the manuscript."

Authors response: We have now performed the biochemical fractions to assess the chromatin bound fraction of 53BP1 in \pm siSET8 cells as suggested by the reviewer. As shown in Fig. R1d, chromatin bound 53BP1 is only moderately affected by the loss of SET8. This is also consistent with the immunofluorescence data shown in Fig. R1c. Since, stable binding of 53BP1 to chromatin (more specifically to sites of damage) requires recognition of both H4K20me2 and H2AK15ub, we argue that loss of SET8 alone do not lead to complete loss of 53BP1 from chromatin.

Fig. R1: SET8-kd does not markedly affect 53BP1 recruitment to DNA damage foci.

a) Design of the experiment. U2OS cells were synchronized by double thymidine block and were reverse transfected with 3 different siRNAs targeting 53BP1. Cells were then either transfected with control or SET8 siRNA during the second block, 6 hours before release from G1/S boundary. Cells were released (T0) and fixed 15 hours from G1/S release (T15). b) Samples from (a) were immunoblotted with the indicated antibodies. * indicates a non-specific protein. c) Cells from (a) were fixed for fluorescence microscopic analysis and stained with the indicated antibodies as well as DAPI for DNA. d) Cells were synchronized and siRNA treated as in (a) and cell pellets were fractionated to obtain soluble proteins and chromatin bound proteins. Samples from both fractions were immunoblotted with the indicated antibodies. * indicates a non-specific protein.

Reviewers' Comments:

Reviewer #2:

Remarks to the Author:

I am generally satisfied with this revision, and the authors are to be commended on a nice study.

Reviewer #3:

Remarks to the Author:

In this revised article, Sorensen and colleagues have addressed my previous points and made the pertinent changes to the text.

My main concern was the apparent lack of coherence between this study and a previous one published in Nat Cell Biol in 2010 (led by one of the current authors). In the rebuttal letter and revised Discussion, the authors strongly argue that the differences can be explained by the different technical approaches used, and they may be correct. If this is the case, the current model derived from this study (Set8 being required to restrict origin licensing in G1 to physiological levels) should replace the previous view of Set8 as a positive regulator of replication origins.

I am satisfied with the revision and think that the article should be published.

REVIEWERS' COMMENTS:

Reviewer #2 (Remarks to the Author):

I am generally satisfied with this revision, and the authors are to be commended on a nice study.

Authors response: We thank the reviewer for their in-depth analysis and efforts to improve our manuscript. We are delighted to learn that the reviewer is satisfied with our responses and the data presented in this study.

Reviewer #3 (Remarks to the Author):

In this revised article, Sorensen and colleagues have addressed my previous points and made the pertinent changes to the text.

My main concern was the apparent lack of coherence between this study and a previous one published in Nat Cell Biol in 2010 (led by one of the current authors). In the rebuttal letter and revised Discussion, the authors strongly argue that the differences can be explained by the different technical approaches used, and they may be correct. If this is the case, the current model derived from this study (Set8 being required to restrict origin licensing in G1 to physiological levels) should replace the previous view of Set8 as a positive regulator of replication origins.

I am satisfied with the revision and think that the article should be published.

Authors response: We appreciate the reviewer for their constructive criticism to improve the manuscript. We are happy that the reviewer is satisfied with the revision.